# Multi-way Encoding for Robustness to Adversarial Attacks

## Abstract

Deep models are state-of-the-art for many computer vision tasks including image classification and object detection. However, it has been shown that deep models are vulnerable to adversarial examples. We highlight how one-hot encoding directly contributes to this vulnerability and propose breaking away from this widely-used, but highly-vulnerable mapping. We demonstrate that by leveraging a different output encoding, multi-way encoding, we can make models more robust. Our approach makes it more difficult for adversaries to find useful gradients for generating adversarial attacks. We present state-of-the-art robustness results for black-box, white-box attacks, and achieve higher clean accuracy on four benchmark datasets: MNIST, CIFAR-10, CIFAR-100, and SVHN when combined with adversarial training. The strength of our approach is also presented in the form of an attack for model watermarking, raising challenges in detecting stolen models.

## 1 Introduction

Deep learning models are vulnerable to adversarial examples [Szegedy et al. (2013)]. Evidence shows that adversarial examples are transferable [Papernot et al. (2016); Liu et al. (2016)]. This weakness can be exploited even if the adversary does not know the target model under attack, posing severe concerns about the security of the models. This is because an adversary can use a substitute model for generating adversarial examples for the target model, also known as *black-box* attacks.

Black-box attacks such as Goodfellow et al. (2014) rely on perturbing input by adding an amount dependent upon the gradient of the loss function with respect to the input of a substitute model. An example adversarial attack is $x^{adv} = x + \epsilon sign(\nabla_x Loss(f(x))$, where $f(x)$ is the model used to generate the attack. This added "noise" can fool a model although it may not be visually evident to a human. The assumption of such gradient-based approaches is that the gradients with respect to the input, of the substitute and target models, are correlated.

Our key observation is that the setup of conventional deep classification frameworks aids in the correlation of such gradients. Typically, a cross-entropy loss, a soft-max layer, and a one-hot vector encoding for a target label are used when training deep models. These conventions make a model more vulnerable to black-box attacks. This setting constrains the encoding length, and the number of possible non-zero gradient directions at the encoding layer. This makes it easier for an adversary to pick a harmful gradient direction and perform an attack.

We aim to increase the adversarial robustness of deep models. Our multi-way encoding representation relaxes the one-hot encoding to a real number encoding, and embeds the encoding in a space that has dimension higher than the number of classes. These encoding methods lead to an increased number of possible gradient directions, as illustrated in Figure 1. This makes it more difficult for an adversary to pick a harmful direction that would cause a misclassification of a correctly classified point, generating a targeted or untargeted attack. Untargeted attacks aim to misclassify a point, while targeted attacks aim to misclassify a point to a specific target class. Multi-way encoding also helps improve a model's robustness in cases where the adversary has full knowledge of the target model under attack: a *white-box* attack. The benefits of multi-way encoding are demonstrated in experiments with four benchmark datasets: MNIST, CIFAR-10, CIFAR-100, and SVHN.

We also demonstrate the strength of our approach by introducing an attack for the recent model watermarking algorithm of Zhang et al. (2018), which deliberately trains a model to misclassify

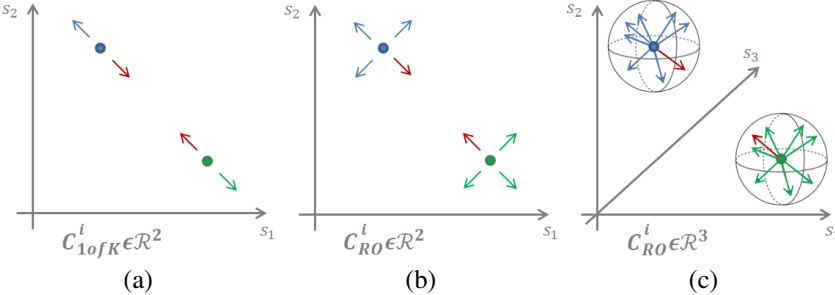

Figure 1: Demonstration of the benefit of relaxing and increasing the encoding dimensionality, for a binary classification problem at the final encoding layer. $C_i$ is the codebook encoding for class $i$, axis $s_i$ represents the output activation of neuron $i$ in the output encoding layer, where $i = 1, \ldots, l$ and $l$ is the encoding dimensionality. The depicted points are correctly classified points of the green and blue classes. The arrows depict the possible non-zero perturbation directions $sign(\frac{\partial Loss}{\partial s_i})$. *(a)* $2D$ $1ofK$ *softmax-crossentropy setup:* Only two non-zero gradient directions exist for a $1ofK$ encoding. Of these two directions, only one is an adversarial direction, depicted in red. *(b)* $2D$ *multi-way encoding:* Four non-zero perturbation directions exist. The fraction of directions that now move a point to the adversarial class (red) drops. *(c)* $3D$ *multi-way encoding:* A higher dimensional encoding results in a significantly lower fraction of gradient perturbations whose direction would move an input from the green ground-truth class to the blue class, or vice versa.

certain watermarked images. We interpret such watermarked images as adversarial examples. We demonstrate that the multi-way encoding reduces the transferability of the watermarked images, making it more challenging to detect stolen models.

We summarize our contributions as follows:

1. We show that the traditional $1ofK$ mapping is a source of vulnerability to adversarial gradients.

2. We propose a novel solution using multi-way encoding to alleviate the vulnerability caused by the $1ofK$ mapping.

3. We empirically show that the proposed approach improves model robustness against both black-box and white-box attacks.

4. We also show how to apply our encoding framework in attacking the recently proposed model watermarking scheme of Zhang et al. (2018).

## 2 RELATED WORK

A wide range of work on adversarial attacks and defenses is presented in Akhtar & Mian (2018). We review recent attacks and defenses that are closely related to our work and present how alternate output encoding schemes have been utilized in deep classification models.

**Attacks.** Adversarial examples are crafted images for fooling a classifier with small perturbations. Recently, many different types of attacks have been proposed to craft adversarial examples. We focus on gradient-based attacks such as [Goodfellow et al. (2014); Kurakin et al. (2016); Athalye et al. (2018)] which deploy the gradient of the loss with respect to the input. Goodfellow et al. (2014) propose the Fast Gradient Sign Method (FGSM) which generates adversarial images by adding the sign of the input gradients scaled by $\epsilon$, where the $\epsilon$ restricts $\ell_\infty$ of the perturbation. Kurakin et al. (2016) propose the Basic Iterative Method (BIM), which is an iterative version of FGSM and is also called Projected Gradient Descent (PGD). Madry et al. (2017) show that PGD with randomly chosen starting points within allowed perturbation can make an attack stronger.

**Defenses.** Most of the state-of-the-art adversarial defenses rely on gradient masking [Papernot et al. (2017)] by designing a defense that makes it more difficult for an adversary to find useful gradients to generate adversarial examples. However, Athalye et al. (2018) show that works including Buckman et al. (2018); Guo et al. (2017); Dhillon et al. (2018); Xie et al. (2017); Song et al. (2017) which

use obfuscated gradients, a special case of gradient masking, are vulnerable to the Backward Pass Differentiable Approximation Attack (BPDA). Defenses that do not use obfuscated gradients, but rely on adversarial training instead [Madry et al. (2017); Kannan et al. (2018)], are robust to BPDA attack. These methods are most similar to our approach because they do not rely on obfuscated gradients. However, Madry et al. (2017) and Kannan et al. (2018) use the conventional one-hot ($1 of K$) encoding for both source and target models, while we propose a higher dimensional multi-way encoding that obstructs the adversarial gradient search.

**Output encoding.** There have been attempts to use alternate output encodings, also known as target encodings, for image classification in deep models. For example, Yang et al. (2015) and Rodríguez et al. (2018) use an output encoding that is based on Error-Correcting Output Codes (ECOC), for increased performance and faster convergence, but not for adversarial defense. In contrast, we use an alternate output encoding scheme, multi-way encoding, to make models more robust to adversarial attacks.

## 3 OUR APPROACH

In this section we will explain our approach using the following notation: $g(x)$ is the target model to be attacked, and $f(x)$ is the substitute model used to generate a black-box attack for $g(x)$. In the case of a white-box attack, $f(x)$ is $g(x)$. Canonical state-of-the-art attacks like FGSM and PGD are gradient-based methods. Such approaches perturb an input $x$ by an amount dependent upon $sign(\nabla_x Loss(f(x)))$. An adversarial example $x^{adv}$ is generated as follows:

$$x^{adv} = x + \epsilon sign(\nabla_x Loss(f(x))), \tag{1}$$

where $\epsilon$ is the strength of the attack. Therefore $x^{adv}$ would be a translated version of $x$, in a vicinity further away from that of the ground-truth class, and thus becomes more likely to be misclassified, resulting in a successful adversarial attack. If the attack is a targeted one, $x$ could be deliberately moved towards some other specific target class. This is conventionally accomplished by using the adversarial class as the ground truth when back-propagating the loss, and subtracting the perturbation from the original input. The assumption being made in such approaches is:

$$\nabla_x Loss(f(x)) \approx \nabla_x Loss(g(x)). \tag{2}$$

We now present the most widely used setup for state-of-the-art deep classification networks. Let the output activation of neuron $i$ in the final encoding (fully-connected) layer be $s_i$, where $i = 1, 2, \ldots, l$ and $l$ is the encoding length. Then, the softmax prediction $y_i$ of $s_i$, and the cross-entropy loss are:

$$y_i = \frac{e^{s_i}}{\sum_{c=1}^k e^{s_c}}, \quad \text{and} \quad Loss = -\sum_{i=1}^k t_i log(y_i), \tag{3}$$

respectively, where $k$ is the number of classes. The partial derivative of the loss with respect to the pre-softmax logit output is:

$$\frac{\partial Loss}{\partial s_i} = y_i - t_i. \tag{4}$$

Combined with the most widely used one-hot ($1 of K$) encoding scheme, the derivative in Eqn. 4 makes the gradients of substitute and target models more correlated. We demonstrate this as follows: Given a ground-truth example belonging to class [1,0,…,0], non-zero gradients of neuron 1 of the encoding layer will always be negative, while all other neurons will always be positive since $0 < y_i < 1$. So, regardless of the model architecture, the signs of the partial derivatives are determined by the category, and thus the gradients for that category only lie in a hyperoctant (see Fig. 1 for the 2D case). This constraint causes strong correlation in gradients in the final layer for different models using the $1 of K$ encoding. Our experiments suggest that this correlation can be carried all the way back to the input perturbations, making these models more vulnerable to attacks.

In this work, we aim to make $\nabla_x Loss(f(x))$ and $\nabla_x Loss(g(x))$ less correlated. We do this by proposing multi-way encoding instead of the conventional $1 of K$ encoding used by deep models for classification. Multi-way encoding significantly reduces the correlation between the gradients of the substitute and target models, making it more challenging for an adversary to create an attack that is able to fool the classification model.

The multi-way encoding we propose in this work is Random Orthogonal ($RO$) output vector encoding generated via Gram-Schmidt orthogonalization. Starting with a random matrix $A = [a_1|a_2|\ldots|a_n] \in \mathbb{R}^{k \times l}$, the first, second, and $k^{th}$ orthogonal vectors are computed as follows:

$$u_1 = a_1, \quad e_1 = \frac{u_1}{||u_1||},$$

$$u_2 = a_2 - (a_2 \cdot e_1)e_1, \quad e_2 = \frac{u_2}{||u_2||}, \qquad\qquad (5)$$

$$u_k = a_k - (a_k \cdot e_1)e_q - \cdots - (a_k \cdot e_{k-1})e_{k-1}, \quad e_k = \frac{u_k}{||u_k||}.$$

For a classification problem of $k$ classes, we create a codebook $C_{RO} \in \mathbb{R}^{k \times l}$, where $C^i = \beta e_i$ is a length $l$ encoding for class $i$, and $i \in 1, \ldots, k$, and $\beta$ is a scaling hyper-parameter dependent upon $l$. A study on the selection of the length $l$ is presented in the experiments section.

By breaking-away from the $1 of K$ encoding, softmax and cross-entropy become ill-suited for the model architecture and training. Instead, we use the loss between the output of the encoding-layer and the $RO$ ground-truth vector, $Loss(f(x), t_{RO})$, where $f(x) \in \mathbb{R}^l$. In our multi-way encoding setup, $s$ and $f(x)$ become equivalent. Classification is performed using $\arg \min_i Loss(f(x), t_{RO}^i)$. We use Mean Squared Error (MSE) Loss.

Figure 1 illustrates how using a multi-way and longer encoding results in an increased number of possible gradient directions, reducing the probability of an adversary selecting a harmful direction that would cause misclassification. For simplicity we consider a binary classifier. Axis $s_i$ in each graph represents the output activation of neuron $i$ in the output encoding layer, where $i = 1, \ldots, l$. The depicted points are correctly classified points for the green and blue classes. The arrows depict the sign of non-zero gradients $\frac{\partial Loss}{\partial s_i}$. (a) Using a $1 of K$ encoding and a softmax-cross entropy classifier, there are only two directions for a point to move, a direct consequence of $1 of K$ encoding together with Eqn. 4. Of these two directions, only one is an adversarial direction, depicted in red. (b) Using 2-dimensional multi-way encoding, we get four possible non-zero gradient directions. The fraction of directions that now move a correctly classified point to the adversarial class is reduced. (c) Using a higher dimension multi-way encoding results in a less constrained gradient space compared to that of $1 of K$ encoding. In the case of attacks formulated following Eqn. 1, this results in $2^l$ possible gradient directions, rather than $l$ in the case of $1 of K$ encoding. The fraction of gradients whose direction would move an input from the green ground-truth class to the blue class, or vice versa, decreases significantly. In addition, multi-way encoding provides additional robustness by increasing the gradients' dimensionality.

We also combine multi-way encoding with adversarial training for added robustness. We use the following formulation to solve the canonical min-max problem [Madry et al. (2017), Kannan et al. (2018)] against PGD attacks:

$$\arg \min_\theta [\mathbb{E}_{(x,y) \in \hat{p}_{data}} (max_{\delta \in S} Loss(\theta, x + \delta, y)) + \lambda \mathbb{E}_{(x,y) \in \hat{p}_{data}} (Loss(\theta, x, y))] \qquad (6)$$

where $\hat{p}_{data}$ is the underlying training data distribution, $(x, y)$ are the training points, and $\lambda$ determines a weight of the loss on clean data together with the adversarial examples at train time.

## 4 EXPERIMENTS

We conduct experiments on four commonly-used benchmark datasets: MNIST, CIFAR-10, CIFAR-100, and SVHN. **MNIST** [LeCun et al. (1998)] is a dataset of handwritten digits. It has a training set of 60K examples, and a test set of 10K examples. **CIFAR-10** [Krizhevsky & Hinton (2009)] is a canonical benchmark for image classification and retrieval, with 60K images from 10 classes. The training set consists of 50K images, and the test set consists of 10K images. **CIFAR-100** [Krizhevsky & Hinton (2009)] is similar to CIFAR-10 in format, but has 100 classes containing 600 images each. Each class has 500 training images and 100 testing images. **SVHN** [Netzer et al. (2011)] is an image dataset for recognizing street view house numbers obtained from Google Street View images. The training set consists of 73K images, and the test set consists of 26K images.

In this work we define a *black-box* attack as one where the adversary knows the architecture but not the weights, and not the output encoding used. This allows us to test the efficacy of our proposed encoding when the adversary assumes the conventional $1 of K$ encoding. We define a *white-box* attack as one where the adversary knows full information about our model, including the encoding.

|  | 10 | 20 | 40 | 80 | 200 | 500 | 1000 | 2000 | 3000 |
|---|---|---|---|---|---|---|---|---|---|
| Black-box | 45.4 | 52.4 | 62.4 | 71.3 | 73.7 | 78.0 | 79.6 | 83.7 | 75.3 |
| Clean | 96.8 | 97.0 | 97.9 | 98.3 | 98.5 | 98.8 | 98.8 | 99.1 | 99.1 |

Table 1: This table presents the effect of increasing the dimension (10, 20, ..., 3000) of the output encoding layer on the classification accuracy (%) of a model that uses $RO$ multi-way encoding for the MNIST dataset on (1) data perturbed using an FGSM black-box attack with $\epsilon = 0.2$ by a model that uses $1ofK$ encoding, and (2) clean data. As the dimension increases, accuracy increases up to a certain point; We use 2000 for the length of our multi-way encoding layer.

| $g(x)$ \ $f(x)$ | $A_{1ofK}$ | $A_{RO}$ | $C_{1ofK}$ | $C_{RO}$ |
|---|---|---|---|---|
| $A_{1ofK}$ | 34.9 (1.00) * | 93.6 (0.02) | 56.8 (0.25) | 95.5 (0.03) |
| $A_{RO}$ | 88.7 (0.02) | 59.1 (1.00) * | 92.5 (0.02) | 83.4 (0.09) |
| $C_{1ofK}$ | 30.1 (0.25) | 84.1 (0.02) | 22.5 (1.00) * | 93.4 (0.01) |
| $C_{RO}$ | 94.3 (0.03) | 87.8 (0.09) | 96.1 (0.01) | 70.5 (1.00) * |

Table 2: This table presents the classification accuracy (%) of MNIST on black-box and white-box FGSM attacks of strength $\epsilon = 0.2$ using architectures A and C. Every cell in this table generates attacks from a substitute model $f(x)$ for a target model $g(x)$. We conclude: a) $g(x)$ is more vulnerable to attacks when $f(x)$ uses the same encoding, hence the lower reported accuracy. b) Even when the source and target models are the same and use the same encoding (*), *i.e.* white-box attacks, $RO$ encoding leads to better accuracy compared to $1ofK$. c) In brackets is the Pearson correlation coefficient of the gradients of $g(x)$ and $f(x)$ with respect to the input $x$. Gradients are less correlated when the source and target models use different encodings. In addition, if the same encoding is used in the source and target models, $RO$ results in a lower correlation compared to $1ofK$.

## 4.1 DEFENSES WITHOUT ADVERSARIAL TRAINING

In this section we analyze the case where neither the target nor substitute model undergoes adversarial training. In all experiments we use $RO$ encoding as the multi-way encoding with dimension 2000 determined by Table 1 and $\beta = 1000$. We first analyze using our multi-way encoding scheme in-depth using the MNIST dataset (4.1.1). We then present results of comprehensive experiments on white-box and black-box attacks, targeted and untargeted, on the four benchmark datasets (4.1.2).

### 4.1.1 MULTI-WAY ENCODING

We conduct experiments to examine how multi-way output encodings can increase adversarial robustness. We compare models trained on $1ofK$ encodings ($A_{1ofK}$ and $C_{1ofK}$) with models having the same architecture but trained on Random Orthogonal output encodings ($A_{RO}$ and $C_{RO}$). Models A and C are LeNet-like CNNs and inherit their names from Tramèr et al. (2017). We use their architecture with dropout before fully-connected layers. We trained models A and C on MNIST with the momentum optimizer and an initial learning rate of 0.01, $momentum = 0.5$. We generated adversarial examples using FGSM with an attack strength $\epsilon = 0.2$. All models achieve $\sim$99% on the clean test set. It should be noted that substitute and target models are trained on clean data and do not undergo any form of adversarial training.

Table 2 presents the classification accuracy (%) of target models under attack from various substitute models. Columns represent the substitute models used to generate adversarial examples and rows represent the target models to be tested on the adversarial examples. The diagonal represents white-box attacks, *i.e.* generating attacks from the target model, and others represent black-box attacks. Every cell in this table generates attacks from a substitute model $f(x)$ for a target model $g(x)$.

It is evident from the results of Table 2 that $g(x)$ is more vulnerable to attacks when $f(x)$ uses the same encoding, hence the lower reported accuracy. This suggests that a model can be far more robust if the output encoding is hidden from an adversary.

It is also evident from the results of this experiment in Table 2 that even when the source and target models are the same, denoted by (*), *i.e.* white-box attacks, and use the same encoding, $RO$

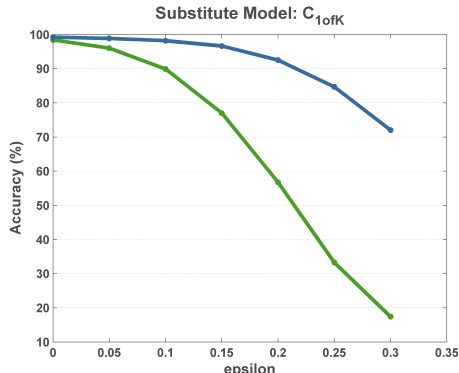 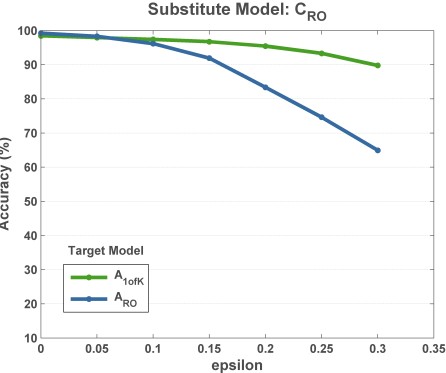

Figure 2: Black-box attacks of varying strength $epsilon$ using $1ofK$ and $RO$ encodings for MNIST. On the left, the substitute model is $C_{1ofK}$, therefore the attacks generated by this model will have a stronger negative effect on a model trained using $1ofK$, and a less negative effect on a model that uses a different output encoding, $RO$. An analogous argument goes for the plot on the right.

encoding leads to better accuracy, and therefore robustness to attack, compared to $1ofK$ encoding. We present further ablation studies in Appendix A.

Finally, Table 2 also reports the Pearson correlation coefficient of $sign(\nabla_x Loss(f(x)))$ and $sign(\nabla_x Loss(g(x)))$ used to perturb an input image $x$ to create an adversarial example $x^{adv}$ as shown in Eqn. 1. These gradients are significantly less correlated when the source and target models use different encodings. In addition, if the same encoding is used in the source and target models, $RO$ results in a lower correlation compared to $1ofK$. We report correlation coefficients for all convolutional layers in Appendix B.

Figure 2 presents black-box FGSM attacks of varying strengths for $1ofK$ and $RO$ encodings. On the left is a $1ofK$ substitute model used to generate attacks for a model originally trained using a $1ofK$ encoding (green), and a model originally trained using a $RO$ encoding (blue). On the right is a $RO$ substitute model used to generate attacks for a model originally trained using a $1ofK$ encoding (green), and a model originally trained using a $RO$ encoding (blue). This confirms that using a different encoding for the source and target models makes the target model more robust to adversarial attacks; Maintaining a higher accuracy even as the strength of the attack increases.

### 4.1.2 BENCHMARK RESULTS

We now demonstrate how using multi-way encoding helps increase robustness in black-box attacks compared to $1ofK$ encoding for both targeted and untargeted attacks on the four benchmark datasets. Targeted attacks are attacks where an adversary would like to misclassify an example to a specific incorrect class. Targeted attacks use the sign of the gradients of the loss on the target class and subtract the perturbation from the original input. We use PGD attacks with a random start, and follow the PGD parameter configuration of Madry et al. (2017), Kannan et al. (2018), and Buckman et al. (2018). Black-box attacks are generated from a substitute model independently trained using a $1ofK$ encoding.

For MNIST and Cifar-10, we follow the experimental settings in Madry et al. (2017); for MNIST we use LeNet, for CIFAR-10 we use a ResNet [He et al. (2016)] of Madry et al. (2017). For Cifar-100 and SVHN we use a WideResNet [Zagoruyko & Komodakis (2016)] of depth 28 and 16, respectively, with a width factor 4 and a dropout of 0.3 following [Buckman et al. (2018)]. We use the optimizer used by Madry et al. (2017) and Buckman et al. (2018).

The result of this experiment is presented in Table 3. In the first column we present the average classification accuracy over all classes for untargeted attacks, and find that models using $RO$ encoding are consistently more resilient to black-box attacks compared to models using $1ofK$ encoding. In the second column we present the average targeted attack success rate over all classes. $RO$ consistently results in a significantly lower attack success rate compared to $1ofK$ for all four benchmark datasets.

| Dataset | Attack | Untargeted Classification Accuracy (%) | Targeted Attack Success Rate (%) |
|---|---|---|---|
| MNIST | Black-box ($1ofK$) | 0.11 | 93.0 |
| | Black-box ($RO$) | **8.7** | **48.4** |
| CIFAR-10 | Black-box ($1ofK$) | 0.5 | 82.3 |
| | Black-box ($RO$) | **5.1** | **50.1** |
| CIFAR-100 | Black-box ($1ofK$) | 5.4 | 13.1 |
| | Black-box ($RO$) | **10.1** | **6.9** |
| SVHN | Black-box ($1ofK$) | 14.5 | 41.5 |
| | Black-box ($RO$) | **34.2** | **21.6** |

Table 3: $RO$ (target model) consistently results in a significantly higher classification accuracy for untargeted attacks, and a significantly lower attack success rate compared to $1ofK$ for all four benchmark datasets. The numbers reported in this table are the average classification and attack success rate over all classes of each dataset. We note that the clean accuracy for MNIST, CIFAR-10, CIFAR-100, and SVHN is, 99.1, 94.3, 74.5, 96.2, respectively ($\pm 0.1$ for $RO$ or $1ofK$).

## 4.2 Defenses with Adversarial Training

In this section we analyze the case where target models undergo adversarial training. This is when adversarial examples are injected in the training data of the target model, making it more difficult for a substitute model to attack. We compare against state-of-the-art methods, which also use adversarial training. All black-box attacks in this section are generated from an independently trained copy of Madry et al. (2017) (substitute model). For adversarial training, we use a mix of clean and adversarial examples for MNIST, CIFAR-10, and CIFAR-100, and adversarial examples only for SVHN following the experimental setup used by Madry et al. (2017) and Buckman et al. (2018).

We compare against state-of-the-art defense methods Madry et al. (2017) and Kannan et al. (2018). Both approaches use a LeNet for MNIST. Madry et al. (2017) presents results for Cifar-10 on a WideResNet (He et al. (2016)), we implement the approach of Kannan et al. (2018) on the same architecture and compare both against our approach. We implement Madry et al. (2017) and Kannan et al. (2018) on WideResNet [Zagoruyko & Komodakis (2016)] following Buckman et al. (2018) and compare against our approach for CIFAR-100 and SVHN.

Table 4 presents the results of combining our multi-way encoding formulation with adversarial training. We obtain state-of-the-art robustness for white-box and black-box attacks, while at the same time increasing the accuracy on the clean dataset for all four benchmark datasets. (*) indicates our replication of Kannan et al. (2018) using the experimental setting of Madry et al. (2017) on MNIST, also used by ours, that uses only $90\%$ of the training set.

## 5 Application: Attacking Model Watermarking

Zhang et al. (2018) introduced an algorithm to detect whether a model is stolen or not. They do so by adding a watermark to sample images of specific classes and deliberately training the model to misclassify these examples to other specific classes. This work has demonstrated to be robust even when the model is fine-tuned on a different training set.

We introduce an attack for this algorithm using our multi-way encoding, making it more challenging to detect whether a model is stolen or not. We do this by fine-tuning the stolen model using multi-way encoding, rather than the encoding used in pre-training the model. We interpret the watermarked image used to deliberately cause a misclassification as an adversarial example. When the encoding of the substitute and target models is different, adversarial examples become less transferable.

We follow the same CIFAR-10 experimental setup for detecting a stolen model as Zhang et al. (2018): We split the test set into two halves. The first half is used to fine-tune pre-trained networks, and the second half is used to evaluate new models. When we fine-tune the $1ofK$ model, we re-initialize the last layer. When we fine-tune the $RO$ model we replace the output encoding layer with our 2000-dimension fully-connected layer, drop the softmax, and freeze convolutional weights.

| Dataset | Attack | Accuracy (%) | | |
|---------|--------|--------------------|----------------------|------|
| | | Madry et al. (2017) | Kannan et al. (2018) | **Ours** |
| MNIST | White-box | 93.2 | 96.4 (93.2*) | **95.4** |
| | Black-box | 96.0 | 97.5 (96.4*) | **97.1** |
| | Clean | 98.5 | 98.8 (98.9*) | **99.0** |
| CIFAR-10 | White-box | 50.0 | 52.9 | **54.1** |
| | Black-box | 64.2 | 66.0 | **67.2** |
| | Clean | 87.3 | 86.8 | **88.5** |
| CIFAR-100 | White-box | 16.2 | 21.5 | **28.5** |
| | Black-box | 38.2 | 42.9 | **45.6** |
| | Clean | 55.3 | 60.1 | **62.5** |
| SVHN | White-box | 41.9 | 46.6 | **49.1** |
| | Black-box | 55.6 | 57.0 | **57.6** |
| | Clean | 90.8 | 90.4 | **91.4** |

Table 4: Comparison against state-of-the-art defense approaches on white-box and black-box PGD attacks, and on clean data. We observe that our approach is more resilient to both types of attacks, while simultaneously improving accuracy on clean data. (*) indicates our replication of Kannan et al. (2018) using the experimental setting of Madry et al. (2017) on MNIST, also used by Ours, that uses only 90% of the training set.

| | Trained from scratch? | Test Accuracy (%) | Watermarking Detection (%) |
|---|---|---|---|
| **StolenNet**$_{1ofK}$ | ✓ | 84.7 | 98.6 |
| **Net**$_{1ofK}$ | ✓ | 48.3 | 6.1 |
| **Net**$_{RO}$ | ✓ | 48.0 | 10.0 |
| **Net**$_{1ofK}$ | fine-tuned from StolenNet | 85.6 | 87.8 |
| **Net**$_{RO}$ | fine-tuned from StolenNet | 80.2 | 12.9 |

Table 5: Our attack is capable of fooling the watermarking detection algorithm. Fine-tuning a stolen model using $RO$ encoding remarkably reduces the watermarking detection accuracy, and makes it comparable to the accuracy of models trained from scratch and do not use the stolen model. The accuracy of fine-tuned models benefits significantly from the pre-trained weights of the stolen model.

We present results on the CIFAR-10 dataset in Table 5. When the fine-tuning was performed using the $1ofK$ encoding (also used in pre-training the model), watermarking detection is 87.8%, and when the fine-tuning was performed using the multi-way $RO$ encoding the watermarking detection is only 12.9%. The watermark detection rate of the model fine-tuned using $RO$ is significantly lower than that fine-tuned using $1ofK$ encoding, and is more comparable to models that are trained from scratch and do not use the stolen model (6.1% and 10.0%). The accuracy of the fine-tuned models benefits significantly from the pre-trained weights of the stolen model.

## 6 CONCLUSION

By relaxing the $1ofK$ encoding to a real number encoding, together with increasing the encoding dimensionality, our multi-way encoding confounds an attacker by making it more difficult to perturb an input in gradient direction(s) that would result in misclassification of a correctly classified example, for a targeted or untargeted attack. We present state-of-the-art results on four benchmark datasets for both black and white-box attacks and achieve higher classification accuracy on clean data. We also demonstrate the strength of our approach by introducing an attack for model watermarking, making it more difficult to detect stolen models.

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

Appendix

# A    ABLATION STUDY ON ENCODINGS

We perform ablation studies to further investigate the effectiveness of our $RO$ encoding. We train the model used in Table 2 with two different combinations of encodings and loss functions.

## A.1    ALTERNATIVE APPROACH

### A.1.1    $RO_{softmax}$

We evaluate a network that uses $RO$ encoding, a softmax layer, and cross-entropy loss. We compute the probability of $i^{th}$ class as follows:

$$P(i|s) = \frac{\exp(\mathbf{s}^\top \mathbf{e_i})}{\sum_{j=1}^{n} \exp(\mathbf{s}^\top \mathbf{e_j})}$$

where $\mathbf{s}$ is the normalized final layer representation, $\mathbf{e_i}$ is the $RO$ encoding vector (ground-truth vector) from the codebook, and $n$ is the number of classes.

### A.1.2    $1ofK_{MSE}$

We also evaluate a network that uses mean-squared error (MSE) loss with the $1ofK$ encoding.

## A.2    EVALUATION

We generate FGSM attacks with $\epsilon = 0.2$ from substitute models $A_{1ofK}$ and $C_{1ofK}$ on MNIST to evaluate the models of Section A.1.1 and Section A.1.2. We also measure a correlation coefficient of the sign of the input gradients between target and substitute models as explained in Section 4.1.1. Tables 6 and 7 demonstrate that $RO$, among the different target models, achieves the highest accuracy and the lowest input gradient correlation with the substitute model.

| Target | A | | | C | | |
|---|---|---|---|---|---|---|
| Model | $RO_{softmax}$ | $1ofK_{MSE}$ | $RO$ | $RO_{softmax}$ | $1ofK_{MSE}$ | $RO$ |
| Accuracy (%) | 48.7 | 43.4 | 88.7 | 53.7 | 42.1 | 94.3 |
| Correlation Coefficient | 0.14 | 0.15 | 0.02 | 0.1 | 0.13 | 0.03 |

Table 6: This table presents black-box attacks from the substitute model $A_{1ofK}$ on various target models. $RO$ achieves the highest accuracy and the lowest input gradient correlation with the substitute model among the different target models.

| Target | A | | | C | | |
|---|---|---|---|---|---|---|
| Model | $RO_{softmax}$ | $1ofK_{MSE}$ | $RO$ | $RO_{softmax}$ | $1ofK_{MSE}$ | $RO$ |
| Accuracy (%) | 67.4 | 55.9 | 92.5 | 62.6 | 58.8 | 96.1 |
| Correlation Coefficient | 0.08 | 0.09 | 0.02 | 0.08 | 0.1 | 0.01 |

Table 7: This table presents black-box attacks from the substitute model $C_{1ofK}$ on various target models. $RO$ achieves the highest accuracy and the lowest input gradient correlation with the substitute model among the different target models.

## B    Correlation of Convolutional layers

In response to the reviewer, we measure the correlation of gradients between all convolutional layers of the different models. We first compute the gradients of the loss with respect to intermediate features of Conv1 and Conv2. Then, we compute the Pearson correlation coefficient of the sign of the gradients with respect to such intermediate features between models. For further comparison, we train models $A'_{1ofK}$ and $A'_{RO}$, that are independently initialized from $A_{1ofK}$ and $A_{RO}$.

| Layer | Input | Conv1 | Conv2 |
|---|---|---|---|
| Correlation Coefficient | 0.35 | 0.29 | 0.25 |

Table 8: Correlation between $A_{1ofK}$ and $A'_{1ofK}$

| Layer | Input | Conv1 | Conv2 |
|---|---|---|---|
| Correlation Coefficient | 0.1 | 0.008 | 0.13 |

Table 9: Correlation between $A_{RO}$ and $A'_{RO}$

| Layer | Input | Conv1 | Conv2 |
|---|---|---|---|
| Correlation Coefficient | 0.02 | 0.005 | 0.01 |

Table 10: Correlation between $A_{RO}$ and $A_{1ofK}$

In order to measure proper correlations, we average gradients of convolutional layers over channels similar to the way used to generate a gradient-based saliency map Selvaraju et al. (2017). Otherwise, the order of convolutional filters affects the correlations and makes it hard to measure proper correlations between models. In this sense, the correlations at FC1 (before the last layer) may not give meaningful information since neurons in the FC layer do not have a strict ordering.

In Table 8 and 9, we find that the correlations of Conv1 and Conv2 between 1ofK models are much higher than those of RO models. In addition, even though RO models used the same output encoding, they are not highly correlated. Table 10 shows that the correlations between RO and 1ofK are also low.

