# OpenReview forum: "Multi-way Encoding for Robustness to Adversarial Attacks"
_ICLR.cc/2019/Conference_

### Official Review · AnonReviewer1 · 2018-11-02
**Promising results, but could use some more experiments**

**Rating:** 6
**Confidence:** 2

**Review:**

This work proposes an alternative loss function to train models robust to adversarial attacks. Specifically, instead of the common sparse, N-way softmax-crossentropy loss, they propose to minimize the MSE to the target column of a random, dense orthogonal matrix. I believe the high-level idea behind this work is that changing the target labelspace is a more effective means of defending against adversarial attacks than modifying the underlying architecture, as the loss-level gradients will be strongly correlated across all architectures in the latter scenario.

Pros:
-Paper was easy to follow
-Using orthogonal encodings to decorrelate gradients is an interesting idea
-Benchmark results appear promising compared to prior works

Cons:
-This work claims that their RO-formulation is fundamentally different from 1-of-K, but I'm not completely sure that's true. One could train a classification model where the final fully connected layer (C inputs K output logits) were a frozen matrix (updates disabled) of K orthogonal basis vector (ie, the same as the C_{RO}) codebook they propose. The inputs to this layer would probably have to be L2 normalized, and the output logits would then proceed through a softmax-crossentropy layer. Would this be any less effective than the proposed scheme?
-Another baseline/sanity test that should probably included is how does the 1-of-K softmax/cross-entropy compare with the proposed method where encoding length l = k and the C_{RO} codebook is just the identity matrix?
-Some of the numbers in Table 4 are pretty close. Since the authors are replicating Kannan et al, it would be best to included error bars when possible to account for differences in random initializations.
-It is unclear the extent to which better classification performance on the clean input generalizes to datasets such as ImageNet

Overall, I think the results are promising, but I'm not fully convinced that similar results cannot be achieved using standard cross-entropy losses with 1-hot labels.

---

> ### Author Response · Authors · 2018-11-22
> **Response to AnonReviewer1**
>
>
>
> Thank you for your review.
>
> 1. and 2. In response to the reviewer’s request, we perform additional experiments with models that we denote as ‘RO_softmax’ (suggestion 1) and ‘1ofK_MSE’ (suggestion 2). We train ‘A_RO_softmax’, ‘A_1ofK_MSE’, ‘C_RO_softmax’, and ‘C_1ofK_MSE’ models, and evaluate the models under FGSM attacks from ‘1ofK’ substitute models. We also report Pearson correlation coefficients of the sign of the input gradients between the substitute and the target models as explained in Section 4.1.1. The results are reported in Appendix A of the manuscript. Models with ‘RO_Softmax’ and ‘1ofK_MSE’ are less robust to attacks from ‘1ofK’ compared to ‘RO’ and have higher correlations between ‘1ofK’ models.
>
> 4. Adversarial robustness on ImageNet is still an open problem.  Even using the conventional 1ofK encoding, it is hard to generalize to clean data when a model is trained with adversarial training [1]. Tsipras et al. adversarially train a model with L-infinity=0.05 on Restricted ImageNet, a smaller subset of ImageNet, and the clean accuracy is decreased by ~70% compared to standard training (without adversarial training).
>
> In addition, adversarial training on ImageNet is very costly.  Kannan et al. [2] managed to perform adversarial training on ImageNet with 53 GPUs. Due to the computational constraints, most of the papers at ICLR 2018 on adversarial robustness do not experiment with ImageNet.  (e.g., Madry et al. ICLR 2018, Buckman et al ICLR 2018, Na et al. ICLR 2018).
>
> We would like to emphasize that scaling-up adversarial training for ImageNet is beyond the scope of this work, but we obtain promising generalizability on smaller scale datasets with adversarial training.
>
> [1] Tsipras, Dimitris, et al. "Robustness may be at odds with accuracy." arXiv preprint arXiv:1805.12152 (2018).
> [2] Kannan, Harini, Alexey Kurakin, and Ian Goodfellow. "Adversarial Logit Pairing." arXiv preprint arXiv:1803.06373(2018).
>
> 3. We present the mean and standard deviation of five runs here:
>
> MNIST: 93.3 (+/- 0.98), 96.5 (+/- 0.31), 98.8 (+/- 0.12)
> SVHN: 45.2 (+/- 1.30), 56.7 (+/- 0.47), 90 (+/- 0.44)
> Cifar100: 21.2 (+/- 0.25), 42.6 (+/- 0.20), 60.0 (+/- 0.27)
> Cifar10: 52.4 (+/- 0.33), 65.8 (+/- 0.13), 86.5 (+/- 0.17)

---

### Official Review · AnonReviewer2 · 2018-11-04
**Novel approach to classification for resiliance against adversial attacks, supported by multiple experiments.**

**Rating:** 6
**Confidence:** 2

**Review:**

This paper argues that a random orthogonal output vector encoding is more robust to adversarial attacks than the ubiquitous softmax. The reasoning is as follows:

1. different models that share the same final softmax layer will have highly correlated gradients in this final layer
2. this correlation can be carried all the way back to the input pertubations
3. the use of a multi-way encoding results in a weaker correlation in gradients between models

I found (2) to be a surprising assumption, but it does seem to be supported by the experiments. These show a lower correlation in input gradients between models when using the proposed RO encoding. They also show an increased resiliance to attack in a number of different settings.

Overall, the results seem to be impressive. However, I think the paper would be a lot stronger if there was a more thorough investigation of the correlation between gradients in all layers of the models. I did not find the discussion around Figure 1 to be very compelling, since it is only relevant to the encoding layer, while we are only interested in gradients at the input layer. The correlation numbers in Table 2 are unexpected and interesting. I would like to see a deeper investigation of these correlations.

I am not familiar with the broader literature in this area, so giving myself low confidence.

---

> ### Author Response · Authors · 2018-11-22
> **Response to AnonReviewer2**
>
>
>
> Thank you for your review.
>
> In response to the reviewer, we measure the correlation of gradients between all convolutional layers of the different models. We first compute the gradients of the loss with respect to intermediate features after Conv1 and Conv2. Then, we compute the Pearson correlation coefficient of the sign of the gradients with respect to such intermediate features between models. For further comparison, we train models ‘A’_1ofK’ and ‘A’_RO’, that are independently initialized from ‘A_1ofK’ and ‘A_RO’. This correlation analysis is reported in Appendix B of the manuscript. We find that the correlations of Conv1 and Conv2 between ‘1ofK’ models are much higher than those of ‘RO’ models. In addition, even though ‘RO’ models used the same output encoding, they are not highly correlated. We also find that the correlations between ‘RO’ and ‘1ofK’ are low.

---

### Official Review · AnonReviewer3 · 2018-11-06
**Review of Multi-way Encoding for Robustness to Adversarial Attacks**

**Rating:** 6
**Confidence:** 3

**Review:**

Authors proposes new method against adversarial attacks. Paper is organized well and easy to follow. Basically, authors notice that gradients of a deep neural network when one hot encoding is used can be highly correlated and hence can be used in the design on an attack. Authors supply extensive experimental evidence to support their method. Those experiments shows significant amount of gains compared to baselines. Although proposed method is neat, I believe it has room to be improved. One question which is bothering me is: Given that one hot encoding is not optimal, can one find optimal (highly resistant to any attack) encoding? One may use evolutionary computing to empirically analyse such a encoding or one may come up an existence/non-existence proof (I am not expert in the field however I guess ecoc field should have investigates similar problems) of such encoding.

---

> ### Author Response · Authors · 2018-11-23
> **Response to AnonReviewer3**
>
>
>
> Thank you for your review.
>
> We agree that optimizing the output encoding to maximize robustness to attack is not sufficiently explored in deep models and is an interesting future direction for our research.

---

### Official Review · AnonReviewer4 · 2018-11-21
**An interesting approach but insufficient evaluation and motivation**

**Rating:** 4
**Confidence:** 4

**Review:**

This paper argues that the vulnerability of classifiers to (black-box) adversarial attacks stems from the use of a final cross-entropy layer trained on one-hot labels. The authors propose replacing this layer by encoding each label as a high-dimensional vector and then training the classifier to minimize the L2 distance of the classifier output from the encoding of the correct class. While the approach is interesting and the paper well-written, both the motivation and the experimental evaluation is insufficient. Hence I consider it below the ICLR bar.

I find the approach weakly motivated. The argument in Figure 1 is very hand-wavy with no clear experimental or theoretical support. The authors argue that cross-entropy with one hot labels causes gradient correlation in the last layer and this propagates all the way through the network (bottom of page 3) but there are no experiments supporting this conjecture.
Moreover, the approach is not fundamentally different from standard networks with cross-entropy training. One can consider adding an extra layer (with number of neurons equal to the encoding vector dimensions) and keeping the weights of these neurons fixed (the output weights are essentially the encoding dictionary). Then training with cross-entropy is increasing the inner product with these vectors. This is qualitatively very similar to the proposed approach of this paper. Is there a benefit from explicitly considering the encoding vectors?

Moreover, why is the length of the encoding vectors important from a conceptual point of view? As far as I can tell, this is simply encouraging the output of the network to be large in norm. This could be leading to gradient masking, similar to the phenomena observed for defensive distillation.

I find the proposed approach to watermark evasion interesting. However I consider it orthogonal to the rest of the results so it is hard to consider it as a contribution to the main point of the paper.

Figure 3 is missing white-box evaluation of RO classifiers. Is this on purpose? It is important to understand if the claimed improvement in robustness actually stems from RO rather than mostly from combining it with adversarial training.

The authors report an increase in white-box adversarial robustness. However, I don't believe that the evaluation of their method is thorough. There are plenty of examples by now where PGD has not been sufficient to evaluate the ground-truth robustness of a model. This can distort the relative robustness of different approaches. Given that the increase from baselines in white-box robustness is relatively small (<10% for most datasets) a much more thorough evaluation is required to conclusively demonstrate the benefit of this method. For instance, applying the SPSA attack from Uesato et al. (2018, https://arxiv.org/abs/1802.05666) or a variant of the CW (https://arxiv.org/abs/1608.04644) attack adapted to the particular method used.
As an additional point of concern emphasizing this issue, the authors present the results of Kannan et al. (2018) as state-of-the-art. So far, there is no conclusive evidence about ALP improving the robustness of neural networks beyond adversarial training. The original paper was found to be not as robust as claimed and retracted from NIPS. A similar paper reporting ALP to improve robustness in smaller datasets (CIFAR10) was submitted to ICLR (https://openreview.net/forum?id=Bylj6oC5K7) but was withdrawn after the authors performed additional experiments. The fact that the authors find the approach of Kannan et al. (2018) to offer an increase over the robustness of Madry et al. (2017) thus raises concerns about the reliability of the evaluation.

Other comments:
-- When the authors perform PGD, what is exactly the loss it is applied on? Is it clear that this is the optimal loss to use when attacking RO classifiers?

---

> ### Author Response · Authors · 2018-11-22
> **Response to AnonReviewer4 (Part 1)**
>
>
>
> Thank you for your review.
>
> 1. In response to the reviewer, we measure the correlation of gradients between all convolutional layers of the different models. We first compute the gradients of the loss with respect to intermediate features of Conv1 and Conv2. Then, we compute the Pearson correlation coefficient of the sign of the gradients with respect to such intermediate features between models. For further comparison, we train models ‘A’_1ofK’ and ‘A’_RO’, that are independently initialized from ‘A_1ofK’ and ‘A_RO’. This correlation analysis is reported in Appendix B of the manuscript. We find that the correlations of Conv1 and Conv2 between ‘1ofK’ models are much higher than those of ‘RO’ models. In addition, even though ‘RO’ models used the same output encoding, they are not highly correlated. We also find that the correlations between ‘RO’ and ‘1ofK’ are low.
>
> In response to the reviewer’s request, we perform additional experiments with a model that we denote as ‘RO_softmax’. We train ‘A_RO_softmax’, ‘C_RO_softmax’ models, and evaluate the models under FGSM attacks from ‘1ofK’ substitute models. We also report Pearson correlation coefficients of the sign of the input gradients between the substitute and the target models as explained in Section 4.1.1. The results are reported in Appendix A of the manuscript. Models with ‘RO_Softmax’ and ‘1ofK_MSE’ are less robust to attacks from ‘1ofK’ compared to ‘RO’ and have higher correlations between ‘1ofK’ models.
>
> 2. Using a higher dimension (>K) results in a less constrained gradient space compared to that of 1ofK encoding. [1] addresses the importance of the norm in the input to the softmax layer (Section 3 of [1]) so that the norm has a massive impact on the softmax output. However, we do not use a softmax and our loss is determined by the relative difference between the output and the ground-truth vector which is not related to the norm of the output. Therefore the explanation in [1] is not directly applicable to our setup.
>
> 3. We demonstrate watermarking evasion to further demonstrate that our proposed approach decorrelates model predictions on watermarked images. We interpret a watermarked image used to deliberately cause a misclassification as an adversarial example. *When the encoding of the substitute and target models is different, adversarial examples become less transferable.*
>
> 4. Table 3 was intended to only show the results of Black-box attacks without adversarial training. The purpose of presenting the White-box (1ofK) result is to show how transferable Black-box attacks (1ofK) are for reference. We remove the row of White-box (1ofK), together with its referenced text, in order to avoid confusion. For the White-box (RO) without adversarial training, we achieve 0.1% classification accuracy on untargeted attacks and 74.7% attack success rate on targeted attacks for MNIST. Without adversarial training, the RO model is still vulnerable to untargeted attacks and shows improvements on targeted attacks. This indicates that our model does not break gradient descent. Considering the space constraints, we decided to instead demonstrate the main argument: RO gives better performance than 1ofK under White-box attacks with adversarial training as reported in Table 4.

---

> > ### Comment · AnonReviewer4 · 2018-11-27
> > **Reviewer's response**
> >
> > I thank the authors for their comments and time spent to perform additional experiments. I do believe that the proposed approach is interesting and has potential. However, I still don't find it mature enough for publication at this point. The evaluation is not as thorough as it could be, while the motivation is still lacking in my opinion. I believe that the paper would benefit significantly from additional work and another round of submission and reviews. Please refer to specific points below.
> >
> > 1. I appreciate the time spent implementing the comparison between MSE and cross-entropy as well as 1ofK vs. RO. I believe that performing such ablation studies is necessary to properly evaluate the merit of a proposed approach. According to the updated results, neither MSE loss or cross-entropy-RO *alone* provide the benefits of the full RO. I would thus argue that the underlying mechanism behind the reported benefits of RO is still not well-understood. The motivation provided does not meaningfully differentiate between these variants that (apparently) have very different performance.
> >
> > 4. From what I understand, there is essentially no white-box (untargeted) robustness achieved by RO alone. White-box robustness is only achieved when RO is combined with adversarial training. I would argue that this significantly weakens the argument towards RO being an important part of the robustness achieved.
> >
> > 5. The authors report an accuracy of ~49% and ~47% against different attacks on CIFAR10. This is considerably below the original reported accuracy of 54%. Given that (according to Table 4) RO improves over state-of-the-art by an amount often close to 5% I would argue that this discrepancy is significant. If we are not able to evaluate the robustness of the model within 5% then we cannot consider a 5% improvement over SOTA significant. I understand that benchmarking against all possible attacks is exhausting and time-consuming. However given how noisy the field of adversarial ML is currently, I believe such scrutiny is necessary to ensure we are making meaningful progress.
> >
> > 6.  When evaluated against SPSA attacks, the concurrent ICLR submission reports numbers worse then PGD training for ALP ( https://openreview.net/forum?id=Bylj6oC5K7&noteId=Hkgxu1t6j7 ).
> >
> > 7. I would like to point out that simply maximizing the MSE loss might not be the optimal attack. To induce misclassification, an adversary needs to move the output of the model to be closer to a wrong class than to a correct class. Maximizing the MSE loss simply drives  the output away from the correct code vector. Hence a variant of the Carlini-Wagner loss where the distance from the correct encoding is maximized while simultaneously minimizing the distance from the closest wrong class would be an important baseline to consider.

---

> > > ### Author Response · Authors · 2018-12-04
> > > **Response to AnonReviewer4 (Round 2)**
> > >
> > >
> > >
> > > Although it is very last minute, we appreciate the interest of the reviewer in our work. As per the reviewer’s request, we ran experiments that utilize the Carlini-Wagner loss and we report the results below in point 7.
> > >
> > >
> > > 1.
> > > We motivate (in Figure 1) and results demonstrate (in Table 1) that a large output dimensionality is key to the performance of RO. In the appendix, we show that RO_softmax and 1ofK_MSE are not as effective as our RO in decorrelating the gradients (see Table 6, 7), and we reveal a very consistent relationship between the gradient decorrelation and the robustness against black box attacks (see Table 6-10). RO_softmax and 1ofK_MSE only have dimension K=10 (vs. 2000 for RO) in the output layer, which limits their gradient space (see Section 3). Note that increasing the output dimension in RO will not encourage an increase of the norm as the RO vectors are on a unit sphere. The benefit of RO comes from a less constrained gradient space leading to better decorrelation of the gradients.
> > >
> > >
> > > 4.
> > > The results in Table 2 directly answer this question. We show that our method achieves added robustness on FGSM white-box attacks (untargeted) in Table 2 without adversarial training, as well as higher robustness than vanilla adversarial training on PGD white-box attacks (untargeted) when combined with adversarial training.
> > >
> > >
> > > 5 and 6.
> > > The 54% accuracy on white-box attacks reported in Table 4 is obtained in the same setting as Madry et al [3], where only 7 iterations are performed for generating the adversarial attacks. The 47.8% accuracy reported in the rebuttal is obtained by increasing the iterations to 1000, which is a much more challenging attack setting than that of Madry et al. We now provide a similar evaluation for Cifar-100 and SVHN on 1000 iterations:
> > >
> > >                      MNIST    Cifar-10    Cifar-100    SVHN
> > > Madry [3]        92.5       45.2           13.4           32.7
> > > Ours                94.2        47.8           27.5           41.4
> > >
> > > The 49% accuracy we reported in the rebuttal is about a query-based attack [5], and 1000 images are tested, following the setting of [9]. We have run the experiment multiple times and find that the results are consistently performing better than Madry et al [3].
> > >
> > > Even the state-of-the-art defense, Madry et al. [3], achieves 0% classification accuracy on MNIST against the other white-box attacks in [10]. Developing a universal defense method for all types of attacks is an open problem and is out of the scope of our paper. We focus on gradient-based attacks and generate the white-box attacks with the same configuration used in [3] for a fair comparison, showing the improved robustness of our RO method under the same settings.
> > >
> > >
> > > 7.
> > > In response to the reviewer’s request, we perform additional experiments on white-box attacks. We evaluate our method on (a) PGD attacks from CW loss and (b) Decision-based attacks [11] (which do not exploit knowledge of the model’s loss function).
> > >
> > > (a) We use a CW loss which minimizes the distance between an output (y) and the ground-truth (t_i) vector while maximizing the distance between the output and the nearest vector (t_j) to the output (y), where i is not equal to j and t_i and t_j are from the codebook. We use the CW loss to train a model and generate PGD attacks. We measure the classification accuracies on 1000-step PGD white-box attacks, black-box attacks from [3], and clean data of Cifar-10.
> > >
> > > 	                          White-box    black-box     clean
> > > Madry et al. [3]             45.2            64.2              87.3
> > > Ours (CW loss)             50.5            69.4              90.9
> > >
> > > (b) We use the Foolbox implementation [12] of the Decision-based attacks, and evaluate 1000 samples on MNIST and set a threshold of 1.5 following [10, 11]. Please note that these attacks do not exploit knowledge of the model’s loss function. We obtain a white-box attack accuracy of 42.7%, while Madry et al. obtain an accuracy of 35.4%.
> > >
> > > We found that our model consistently performs better than [3] on the PGD attacks from CW loss and the Decision-based attacks. These attacks do not weaken our claim that RO encoding performs better than 1ofK encoding.
> > >
> > >
> > >
> > > [3] Madry, A., Makelov, A., Schmidt, L., Tsipras, D., and Vladu, A. Towards deep learning models resistant to adversarial attacks. ICLR, 2018.
> > > [10] Schott, Lukas, et al. "Towards the first adversarially robust neural network model on MNIST." CoRR, abs/1805.09190(2018). (https://arxiv.org/abs/1805.09190)
> > > [11] Brendel, Wieland, Jonas Rauber, and Matthias Bethge. "Decision-based adversarial attacks: Reliable attacks against black-box machine learning models." ICLR 2018
> > > [12] Rauber, Jonas, Wieland Brendel, and Matthias Bethge. "Foolbox v0. 8.0: A python toolbox to benchmark the robustness of machine learning models." arXiv preprint arXiv:1707.04131(2017).

---

> > > > ### Comment · AnonReviewer4 · 2018-12-04
> > > > **The authors appear to be confusing the notions of a threat model and an attack configuration.**
> > > >
> > > > In this research area, the metric we are interested in is "robust (or adversarial) accuracy given a threat model". This is defined as the _minimum_ accuracy achieved against _any_ attack performed within the threat model limits. Hence if the model achieves 54% accuracy against 7-step PGD and 47% accuracy against 1000-step PGD, then the models _robust accuracy_ is 47% (or lower if better attacks exist). The goal is not to produce models that are resistant to existing attack but rather models that are resistant to _all_ attacks within a threat model. Here, threat model refers to the space of allowed adversarial perturbations (e.g. at most 0.3 in Linfinity norm).
> > > >
> > > > The authors mention that the Madry et al. models have 0% accuracy against different attacks. This is not true. The attacks evaluated in [10] _violate the threat model_ (this is intentional and properly explained in [10]).

---

> > > > > ### Author Response · Authors · 2018-12-08
> > > > > **Response to AnonReviewer4 (Round 3)**
> > > > >
> > > > >
> > > > >
> > > > > We thank the reviewer for their clarification as we had misinterpreted “all possible attacks” in the reviewer’s second comment.
> > > > >
> > > > > We agree with the reviewer that evaluating a defense method against all possible attacks under a threat model and reporting the minimum accuracy is a very strong and important setting. Comparisons indicate that our method continues to perform better than state-of-the-art, as we approach the lower bound (minimum). We can further clarify this point in the final version of the paper.
> > > > >
> > > > > However, the goals of a defense method also include making an adversary computationally inefficient. Therefore, a defense that makes low-cost attacks ineffective is also valuable. In terms of the computational cost, the SPSA attacks are computationally much more expensive than the PGD attacks in order to estimate gradients. The SPSA attack involves at most ~2.4 *10^6 -step (batch size = 8192 * iteration = 300) forward passes per image and the n-step PGD attack involves n iterations of backward passes per image. In this work, we demonstrate the benefits of our proposed target encoding, RO, over the conventional 1ofK encoding. This includes added robustness against low-cost attacks over 1ofK encoding.

---

> ### Author Response · Authors · 2018-11-22
> **Response to AnonReviewer4 (Part 2)**
>
>
>
> 5. In response to the reviewer’s request, we test query-based attacks. Since we use a different output encoding and loss function, the existing implementations of SPSA and the Decision Attack cannot be directly applied to our model for technical reasons (library wrappers). Considering the time we need to modify the implementation and validate the code, we instead use another publicly available query-based attack [5], which are more powerful than [2, 6]. Due to the time complexity (at most 1.2x10^5 forward passes per image), we tried the first 1000 images in the test set for Cifar-10. Please note that [9] also perform their attacks on 1000 images of Cifar-10 for evaluation. We obtain an accuracy of 49.10%, while the pre-trained model of [3] obtain an accuracy of 45.6%.
>
> We also demonstrate robustness when we increase the number of iterations used to generate PGD white-box attacks to 100 and 1000. Here, we report our classification accuracy and compare it to results from the publicly released model of Madry et al. [3]. For MNIST, we obtain an accuracy of 94.47% and 94.21%, while Madry et al. [3] obtain an accuracy of 92.53% and 92.45% at 100 and 1000 iterations, respectively. For Cifar-10, we obtain an accuracy of 47.94% and 47.80%, while Madry et al. obtain an accuracy of 45.35% and 45.23% at 100 and 1000 iterations, respectively. We observe that the increase in the order of magnitude of the number of iterations used to generate the attack does not significantly impact our classification accuracy, and maintains a higher accuracy compared to Madry et al. [3]. The results we report in the submission use 7 iterations for Cifar-10 and 40 iterations for MNIST, which was also used in reporting the results in Madry et al. [3], Buckman et al. [7], and Kannan et al. [8].
>
> In addition, we validate that iterative attacks perform better than one-step attacks, white-box attacks are more powerful than black-box attacks, and unbounded attacks achieve 100% success, all of which are desirable properties of defenses in regard to obfuscated gradients [2].
>
> 6. At the time of submission, the ALP [8] paper had not been retracted yet. We do not know clear reasons why they retracted the paper but ALP increases robustness against gradient-based white-box attacks over [3]. The work mentioned by the reviewer (https://openreview.net/forum?id=Bylj6oC5K7) shows that ALP achieves higher accuracy against 1000-step PGD attacks compared to [3] (ALP:48.34% vs. Madry et al: 45.15%). In addition, Mosbach et al. [4] also show added robustness for ALP against gradient-based attacks over [3]. Our evaluation is consistent with these observations.
>
> Aside from ALP, as stated in our original manuscript, our method performs better than [3]. We are happy to omit the comparison to ALP if it is questionable.
>
> 7. We use the same loss function, Mean Squared Error loss, for both training a model and generating adversarial examples following works using gradient-based attacks [3,7,8].
>
> [1] Carlini, Nicholas, and David Wagner. "Defensive distillation is not robust to adversarial examples." arXiv preprint arXiv:1607.04311 (2016).
>
> [2] Athalye, Anish, Nicholas Carlini, and David Wagner. "Obfuscated gradients give a false sense of security: Circumventing defenses to adversarial examples." arXiv preprint arXiv:1802.00420(2018).
>
> [3] Madry, A., Makelov, A., Schmidt, L., Tsipras, D., and Vladu, A. Towards deep learning models resistant to adversarial attacks. ICLR, 2018.
>
> [4] Mosbach, Marius, et al. "Logit Pairing Methods Can Fool Gradient-Based Attacks." arXiv preprint arXiv:1810.12042(2018).
>
> [5] NATTACK: A Strong and Universal Gaussian Black-Box Adversarial Attack, ICLR 2019 submission.
>
> [6] Ilyas, Andrew, et al. "Black-box Adversarial Attacks with Limited Queries and Information." arXiv preprint arXiv:1804.08598(2018).
>
> [7] Buckman, J., Roy, A., Raffel, C., and Goodfellow, I. Thermometer encoding: One hot way to resist adversarial examples. ICLR, 2018.
>
> [8] Kannan, H., Kurakin, A., and Goodfellow, I. Adversarial Logit Pairing. arXiv preprint arXiv:1803.06373.
>
> [9] Uesato, Jonathan, et al. "Adversarial risk and the dangers of evaluating against weak attacks." arXiv preprint arXiv:1802.05666 (2018).

---

### Public Comment · (anonymous) · 2018-10-20
**More Experiments Needed**

I am not convinced that using multi-way encoding as target code will make model more robust. It seems to me this is just another case where gradient become less robust for finding adversarial images, especially when the code length is much larger than the number of classes. You should try increasing the number of iterations for your attack methods, something like increasing from 10 iterations to 100 or 1000 iterations. I have a feeling that this will seriously decrease the performance and make it the same as before.

---

> ### Author Response · Authors · 2018-10-23
> **Results**
>
> As suggested, we set the number of iterations used to generate PGD white-box attacks to 100 and 1000. Here, we report our classification accuracy and compare it to results from the publicly released model of Madry et al. [1]. For MNIST, we obtain an accuracy of 94.47% and 94.21%, while Madry et al. [1] obtain an accuracy of 92.53% and 92.45% at 100 and 1000 iterations, respectively. For Cifar-10, we obtain an accuracy of 47.94% and 47.80%, while Madry et al. obtain an accuracy of 45.35% and 45.23% at 100 and 1000 iterations, respectively. We observe that the increase in the order of magnitude of the number of iterations used to generate the attack does not significantly impact our classification accuracy, and maintains a higher accuracy compared to Madry et al. [1]. The results we report in the submission use 7 iterations for Cifar-10 and 40 iterations for MNIST, which was also used in reporting the results in Madry et al. [1], Buckman et al. [2], and Kannan et al. [3].
>
> [1] Madry, A., Makelov, A., Schmidt, L., Tsipras, D., and Vladu, A. Towards deep learning models resistant to adversarial attacks. ICLR, 2018.
>
> [2] Buckman, J., Roy, A., Raffel, C., and Goodfellow, I. Thermometer encoding: One hot way to resist adversarial examples. ICLR, 2018.
>
> [3] Kannan, H., Kurakin, A., and Goodfellow, I. Adversarial Logit Pairing. arXiv preprint arXiv:1803.06373.

---

### Public Comment · (anonymous) · 2018-10-26
**Table 3 White-Box (RO) missing**

In Table 3, The White-Box (RO) column is not present. Is there a reason for this omission?

---

> ### Author Response · Authors · 2018-11-22
> **Response**
>
>
>
> Thank you for your review.
>
> Table 3 was intended to only show the results of Black-box attacks without adversarial training. The purpose of presenting the White-box (1ofK) result is to show how transferable Black-box attacks (1ofK) are for reference. We remove the row of White-box (1ofK), together with its referenced text, in order to avoid confusion.
>
> For the White-box (RO) without adversarial training, we achieve 0.1% classification accuracy on untargeted attacks and 74.7% attack success rate on targeted attacks for MNIST. Without adversarial training, the RO model is still vulnerable to untargeted attacks and shows improvements on targeted attacks. However, the main argument is that RO gives better performance than 1ofK under White-box attacks with adversarial training as reported in Table 4.

---

### Public Comment · (anonymous) · 2018-11-14
**No black-box query attacks were tried**

This paper makes several black-box claims but no attacks that query the model were tried (e.g., the Decision Attack from ICLR'18 or SPSA from Uesato et al. 2018 at ICML'18). Could the authors try either of these attacks?

---

> ### Author Response · Authors · 2018-11-22
> **Response**
>
>
>
> Thank you for your review.
>
> Like [1] and [2], we did not consider the threat model where an adversary has the ability to send queries to the target model. Since our submission shows the vulnerability between models when the same 1ofK encoding is used and introduces a new output encoding, the black-box threat model in our paper is designed to test whether it is useful to hide the output encoding (codebook) from the adversary. To be specific, our experiments are designed to test the efficacy of our proposed encoding when the adversary assumes the conventional 1ofK encoding is used. Therefore, we are more interested in how the black-box attacks from a model with 1ofk encoding are transferable to the model with our proposed encoding. In this context, we also present the watermarking attack that further demonstrates how adversarial examples are transferable when different output encodings are used.
>
> In response to the reviewer’s request, we test query-based attacks. Since we use a different output encoding and loss function, the existing implementations of SPSA and the Decision Attack cannot be directly applied to our model for technical reasons (library wrappers). Considering the time we need to modify the implementation and validate the code, we instead use another publicly available query-based attack [3], which are more powerful than [5, 6]. Due to the time complexity (at most 1.2x10^5 forward passes per image), we tried the first 1000 images in the test set for Cifar-10. Please note that [4] also perform their attacks on 1000 images of Cifar-10 for evaluation. We obtain an accuracy of 49.10%, while the pre-trained model of [1] obtain an accuracy of 45.6%.
>
> [1] Madry, Aleksander, et al. "Towards deep learning models resistant to adversarial attacks." ICLR 2018.
>
> [2] Kannan, Harini, Alexey Kurakin, and Ian Goodfellow. "Adversarial Logit Pairing." arXiv preprint arXiv:1803.06373(2018).
>
> [3] NATTACK: A Strong and Universal Gaussian Black-Box Adversarial Attack, ICLR 2019 submission.
>
> [4] Uesato, Jonathan, et al. "Adversarial risk and the dangers of evaluating against weak attacks." arXiv preprint arXiv:1802.05666 (2018).
>
> [5] Athalye, Anish, Nicholas Carlini, and David Wagner. "Obfuscated gradients give a false sense of security: Circumventing defenses to adversarial examples." arXiv preprint arXiv:1802.00420(2018).
>
> [6] Ilyas, Andrew, et al. "Black-box Adversarial Attacks with Limited Queries and Information." arXiv preprint arXiv:1804.08598(2018).

---

### Public Comment · (anonymous) · 2018-11-14
**Do not fully understand method**

I am trying to reproduce the results of this paper but I do not fully understand how the proposed method is intended to be trained and classify an input.

The paper says "Instead, we use the loss between the output of the encoding-layer and the RO ground-truth vector". How do we compute this loss?

Specifically, what method is used to compute Loss(f(x), t_RO)?

Further, the paper never defines t_RO? What is it equal to?

Finally, how should \beta be selected?

Would the authors be willing to release source code?

---

> ### Author Response · Authors · 2018-11-22
> **Response**
>
>
>
> Thank you for your review.
>
> 1. This can be any loss function that measures a distance between the vectors. We use the Mean Squared Error Loss as described in the paper.
>
> 2. t_RO is the RO ground-truth vector for a class in the C_RO codebook.
>
> 3. \beta is empirically determined to be 1000 as stated in Section 4.1.
>
> 4. Yes, we are working on releasing the code.

---

### Public Comment · (anonymous) · 2019-04-29
**Code repository**

Do you plan to release your code?

---

### Meta-Review · Area_Chair1 · 2018-12-16
**Interesting idea but the claims need to be still justified better**

**Confidence:** 5
**Recommendation:** Reject

**Metareview:**

This paper proposes a method for improving robustness to black-box adversarial attacks by replacing the cross-entropy layer with an output vector encoding scheme. The paper is well-written, and the approach appears to be novel. However, Reviewer 4 raises very relevant concerns regarding the experimental evaluation of the method, including (a) lack of robustness without AT in the whitebox case (which is very relevant as we still lack good understanding of blackbox vs whitebox robustness) (b) comparison with Kannan et al and (c) lack of some common strong attacks. Reviewer 1 echoes many of these concerns.